# Spitz Tumors and Melanoma in the Genomic Age: A Retrospective Look at Ackerman’s Conundrum

**DOI:** 10.3390/cancers15245834

**Published:** 2023-12-14

**Authors:** Carmelo Urso

**Affiliations:** Dermatopathology Study Center of Florence, I-50129 Florence, Italy; cylaur@libero.it

**Keywords:** Spitz nevus, atypical Spitz tumor, Spitz melanoma, Spitzoid neoplasms, melanoma

## Abstract

**Simple Summary:**

The differential diagnosis between Spitz nevus and Spitz melanoma is challenging. The invoked reason for this diagnostic problem is the close similarity of these lesions. Genetic studies suggest that these tumors, both characterized by large spindle and/or epithelioid cells, are similar, because they belong to the same class of neoplasms. They have similar morphological and phenotypic characteristics, because they harbor the same genetic drivers, and exhibit different biological behaviors, because they possess different mutation burdens. Rather than two opposite groups of benign and malignant neoplasms (Spitz nevus-Spitz melanoma), Spitz tumors appear to form a genomic and bio-morphological spectrum. A ternary classification Spitz nevus-Spitz melanocytoma-Spitz melanoma appears to be more adherent to the actual oncogenetic process, but a fraction of cases remain difficult to diagnose and of uncertain clinical behavior. A prognostic stratification, based on the integration of morphological and genomic features, could be a useful complement to the diagnosis.

**Abstract:**

After 25 years, “Ackerman’s conundrum”, namely, the distinction of benign from malignant Spitz neoplasms, remains challenging. Genomic studies have shown that most Spitz tumors harbor tyrosine and serine/threonine kinase fusions, including *ALK*, *ROS1*, *NTRK1*, *NTRK2*, *NTRK3*, *BRAF* and *MAP3K8*, or some mutations, such as *HRAS* and *MAP3K8*. These chromosomal abnormalities act as drivers, initiating the oncogenetic process and conferring basic bio-morphological features. Most Spitz tumors show no additional genomic alterations or few ones; others harbor a variable number of mutations, capable of conferring characteristics related to clinical behavior, including *CDKN2A* deletion and *TERT*-p mutation. Since the accumulation of mutations is gradual and progressive, tumors appear to form a bio-morphologic spectrum, in which they show a progressive increase of clinical risk and histological atypia. In this context, a binary classification Spitz nevus-melanoma appears as no longer adequate, not corresponding to the real genomic substrate of lesions. A ternary classification Spitz nevus-Spitz melanocytoma-Spitz melanoma is more adherent to the real neoplastic pathway, but some cases with intermediate ambiguous features remain difficult to diagnose. A prognostic stratification of Spitz tumors, based on the morphologic and genomic characteristics, as a complement to the diagnosis, may contribute to better treatment plans for patients.

## 1. Introduction

In 1999, A. Bernard Ackerman wrote that, in melanocytic pathology, of all the diagnostic pitfalls, none was “more grievous” than mistaking a malignant melanoma for a nevus, and of all problematic nevi, none was “more vexing” than Spitz nevus. The “obvious reason” for this diagnostic “conundrum” was that Spitz nevus had more histopathological features in common with melanoma than attributes different from it. They were two formidable mutual simulators: Spitz nevus simulated malignant melanoma and malignant melanoma simulated Spitz nevus. However, despite the many findings shared, Spitz nevus and melanoma, like two “identical twins”, could be distinguished from one another if some “crucial characteristics” were accurately considered. In fact, the criteria for differentiating Spitz nevus from melanoma “worked” and “errors in diagnosis” resulted “not from limitations in the criteria, but from failings in brains that apply them” [1].

As a matter of fact, large spindle and epithelioid neoplasms are highly controversial skin melanocytic tumors [2,3]. Because of their histological atypia, they were originally regarded as melanomas with a favorable prognosis, occurring in childhood (*juvenile melanomas*) [4]. However, shortly afterwards, because of their benign clinical behavior, *juvenile melanomas* were classified among benign nevi [5]. A few years later, *juvenile melanomas*, found to occur also in adults, were renamed as *spindle cell and epithelioid cell nevi* [6,7] and next as *Spitz nevi* [8]. Because Spitz nevi were entirely benign lesions, cases with a subsequent unfavorable outcome were considered as diagnostic errors [6]. However, some malignant tumors showing the histological features of Spitz nevus were proved to exist [9]. These tumors were interpreted as malignant melanomas resembling or mimicking benign Spitz nevi, laying the foundations of the theory of the double mutual simulation Spitz nevus-melanoma [1]. Moreover, subsequently, a small number of lesions showing the histological features of Spitz nevi and lymph node metastases were reported. Such particular cases were not regarded as diagnostic errors and did not seem to be true melanomas, because their histological features were not sufficient for such a diagnosis and because they did not show “the potential of widespread metastasis” [10]. These lesions, first termed as *malignant Spitz nevi*, were later regarded as melanocytic tumors intermediate between benign Spitz nevus and malignant melanoma and labeled as *atypical Spitz nevi/tumors* [11,12]. The concept of borderline lesion appeared to be useful to classify problematic Spitz neoplasms, but Dr. Ackerman refused it, regarding lesions labeled as atypical Spitz tumors only a confused assemblage of morphologically ambiguous benign Spitz nevi and morphologically ambiguous malignant melanomas, because only two diagnoses were possible, “Spitz nevus” and “malignant melanoma” [2].

After 25 years and many advancements in genetic studies, a re-evaluation of Ackerman’s conundrum may contribute to better understand its real reasons and large spindle and epithelioid melanocytic neoplasms.

## 2. Genetic Drivers in Spitz Tumors

Spitz tumors harbor some chromosomal abnormalities, interpreted as driver events, that is, genetic alterations that initiate the oncogenetic process and contribute to confer peculiar morphological features, including the distinctive large spindle/epithelioid cells that characterize such melanocytic tumors [13,14] (Table 1).

Many Spitz tumors show genomic rearrangements, consisting of chromosomal translocations with some consequent tyrosine kinase fusions. These aberrations result in activation of various signaling pathways influencing cell proliferation, increasing cell size and survival, such as the RAS-ERK, RAS-RAF-MEK1/2-ERK1/2, MAPK/ERK, JAK3-STAT3, PI3K-AKT-mTOR, PLC-γ1 and β-catenin pathways [15,16]. They include *ALK*, *ROS1*, *NTRK1*, *NTRK2*, *NTRK3*, *RET* and *MET* kinase fusions [15,16,17]. *ALK* (anaplastic lymphoma kinase) is a gene located on chromosome 2 (2p23), encoding a tyrosine kinase receptor [18]; cases showing *ALK* fusions approximately account for 10% [15]; several partner genes have been detected, including *TPM3* (tropomyosin 3—location 1q21), *DCTN1* (dynactin subunit 1—location 2p13), and *MLPH* (melanophilin—location 2q37) [15,18]. *ROS1* (proto-oncogene tyrosine-protein kinase ROS) is proto-oncogene situated on chromosome 6 (6q22) that encodes another tyrosine kinase receptor [18]; *ROS1* fusion is approximately found in about 15–17% of cases [15]; many fusion partner genes have been detected, including *PWWP2A* (PWWP domain containing 2A—5q33) and *TPM3* [15,18]. *NTRK* (neurotropic receptor tyrosine kinase) genes encode a group of tyrosine kinase receptors. *NTRK1* is located on chromosome 1 (1q23), *NTRK2* on chromosome 9 (9q21) and *NTRK3* on chromosome 15 (15q25) [18]; several partner genes have been reported, including *LMNA* (laminin A/C—location 1q22), *TP53* (tumor protein 53—location 17p13), *TPM3*, *ETV6* (ets variant 6—location 12p13.2), *MYO5A* (myosin VA—location 15q21.2) and *MYH9* (myosin, heavy polypeptide 9, non-muscle—location 22q12.3) [15,18,19,20]. *RET* (rearranged during transfection) is a proto-oncogene, positioned on chromosome 10 (10q11), encoding for a transmembrane glycoprotein receptor with tyrosine kinase activity [18]; *RET* fusion is found in about 5% of tumors [15]; fusion partner genes include *GOLGA5* (golgin A5—location 14q32) and *KIF5B* (kinesin family member 58, location 10p11) [15,18]. The proto-oncogene *MET* (met proto-oncogene), located on chromosome 7 (7q31), encodes a protein, hepatocyte growth factor (HGF) receptor, also showing a tyrosine kinase activity [18]; *MET* fusions are rarely detected [16].

A fraction of Spitz tumors show different genomic rearrangements, consisting of chromosomal translocations associated with serine/threonine kinase fusions. These alterations produce an increased kinase enzymatic activity with activation of MEK1/2 and ERK1/2 signaling pathways, influencing cell proliferation, division and differentiation [15]. They include *BRAF* and *MAP3K8* kinase fusions [14,15,21,22]. The proto-oncogene *BRAF* (v-Raf murine sarcoma viral oncogene homolog B1), encoding a serine/threonine kinase with three different domains (CR1, CR2 and CR3), is found on chromosome 7 (7q34) [18]; *BRAF* fusion is found in approximately 5% of cases [15]; fusion partner genes include *CEP89* (centrosomal protein 89—location 19q13), *LSM14A* (LSM14A mRNA processing body assembly factor—location 19q13), *AKAP9* (A kinase (PRKA) anchor protein (yotiao) 9—location 7q21) and *MAD1L1* (mitotic arrest deficient 1 like 1—location 7p22) [15,18,22]. *MAP3K8* (mitogen-activated protein kinase kinase kinase 8) is a gene located on chromosome 10 (10p11) that encodes a serine/threonine and tyrosine kinase that activates ERK1/2 through phosphorylation of its direct substrate MEK [18,22]; numerous fusion partner genes have been reported, including *SVIL* (supervillin—location 10p11), *UBL3* (ubiquitin like 3—location 13q12), *SPECC1* (sperm antigen HCMOGT-1—location 17p11), *STX7* (syntaxin 7—location 6q23), *GNG2* (G protein subunit gamma 2—location 14q22), *PRKACB* (protein kinase cAMP-activated catalytic subunit beta—location 1p31), *DIP2C* (disco interacting protein 2 homolog C—location 10p15) and CUBN (cubilin—location 10q13) [18,22]. *MAP3K8* and *ALK* fusions seem to be the most frequent drivers in tumors affecting pediatric patients (<18 years old) [23].

More rarely, other fusions may be observed, involving various genes, including *ARAF* (A-Raf proto-oncogene serine/threonine kinase), located on chromosome X (Xp11) [18,22]; *MITF* (micropthalmia-associated transcription factor), located on chromosome 3 (3p14) [18,22]; *PRKCA* (protein kinase C alpha), located on chromosome 17 (17q24) [18,22]; *PRKCB* (protein kinase C beta), located on chromosome 16 (16p12) [18,22]; *RASGRF1* (Ras protein-specific guanine nucleotide-releasing factor 1), located on chromosome 15 (15q25) [18,24,25]; *PRKDC* (protein kinase DNA-activated catalytic subunit), located on chromosome 8 (8q11) [18,25]; *FGFR1* (fibroblast growth factor receptor 1), located on chromosome 8 (8p11) [18,24]; *ERBB4* (erb-B2 receptor tyrosine kinase 4), located on chromosome 2 (2q34) [18,24]; *RAF1* (RAF proto-oncogene serine/threonine-protein kinase), located on chromosome 3 (3p25) [18,24]; *MAP3K3* (mitogen-activated protein kinase kinase kinase 3), located on chromosome 17 (17q23) [18,24]; and *MERTK* (MER proto-oncogene tyrosine kinase), located on chromosome 2 (2q13) [18,26].

Moreover, a proportion of Spitz tumors show some distinctive mutations. The most common is a mutation of *HRAS* (Harvey rat sarcoma viral oncogene homolog), a proto-oncogene located on chromosome 11 (11p15) [18,27]. *HRAS* mutation may be associated with isolated gains of the p-arm of chromosome 11 and *HRAS* gene amplification [28]. Such genetic alterations produce an activation and deviating expressions of factors involved in cell growth and differentiation, via the MAP/ERK and the PI3K-AKT-mTOR signaling pathways [14,27,28]. Some tumors show other less frequent mutations, involving *MAP3K8* [22,24] or *MAP2K1* genes [29,30]. *MAP3K8* truncation may lead to an increased activity of the MEK1/2 and ERK1/2 pathways, influencing cell proliferation and differentiation [22,24]. *MAP2K1* (dual specificity mitogen-activated protein kinase kinase 1) is a proto-oncogene located on chromosome 15 (15q22), encoding MEK1, a serine/threonine and tyrosine kinase, involved in the RAF-MEK1/2-ERK1/2 pathway [18,29]. More frequently, the activating mutation in *MAP2K1* is an in-frame deletion [29], more rarely, a missense mutation [30]. The frequency of these genomic alterations seems to vary depending on the anatomic location of tumors. *ALK* and *ROS1* fusions are frequently found in tumors of the lower extremities; *NTRK* and *BRAF* fusions are common in lesions of lower and upper extremities; *HRAS* mutation is more frequently found in tumors located on the head-neck region and the extremities; *MAP3K8* fusion/truncation has been found in lesions of the extremities, trunk and head-neck region; *MAP2K1* mutation is found in tumors located on the extremities [29,31,32].

In addition to these genomic abnormalities, it is reasonably certain that Spitz tumors present other genetic drivers, probably less frequent, yet to be discovered. Recently, a novel three-way complex rearrangement has been reported [33]. The involved genes were *TRPM1* (transient receptor potential cation channel, subfamily M, member 1—location 15q13) and *LCK* (lymphocytic-specific protein tyrosine kinase—location 1p35.1), together with *PUM1* (pumilio RNA binding family member 1—location 1p35.2), a gene placed very close to the latter [18,33]. Moreover, some lesions morphologically overlapping to Spitz tumors and harboring *BRAF*, most frequently V600E, *NRAS* (neuroblastoma RAS viral oncogene homolog—location 1p23) or *NF1* (neuro-fibromin 1—location 17q11) mutations, have been found [18,24,34]. Because such mutations frequently have the role of drivers in some conventional nevi and melanomas, including superficial spreading and lentigo maligna melanomas [35,36,37,38], tumors showing *BRAF*, *NRAS* or *NF1* mutations are no longer considered true Spitz tumors, but simulators or mimickers of them [37]. For tumors showing a *BRAF*^V600E^ mutation, a provisional label of BAMS (BRAF-mutated and morphologically Spitzoid) tumors has also been provided [34].

## 3. Other Genomic Aberrations in Spitz Tumors

Driver fusions/mutations, generally mutually exclusive, induce an increase of melanocytic proliferation leading to a clonal expansion of partially transformed cells [38]. In this initial phase, the melanocytic proliferation is limited by several cellular mechanisms that regulate cell growth, inducing senescence and apoptosis [14]. However, the augmented proliferative activity increases the likelihood that other mutations occur. These additional genetic alterations may be ineffective (passenger mutations) or capable of modifying, slightly or severely, one or more of the cellular mechanisms that normally restrain unlimited growth [38,39,40]. Such modifications may result in an enhancement of cell proliferation and in a further increase in the probability that further mutations take place [38,39,40]. In turn, these supervening mutations may be capable of activating additional pro-survival pathways, of suppressing pro-apoptotic pathways, of ablating important tumor-suppression mechanisms and of disrupting DNA repair systems [14,38,39,40]. Subsequently, the oncogenetic process may continue with the acquisition of additional complex characteristics related to the clinical behavior of cells, including the capacity for local invasion, widely spreading, evading immune destruction, induction of angiogenesis, colonization and growth in distant organs [13,14,38,39,40]. Possible additional mutations may involve several genes. *TERT*-promoter mutation can avoid shortening of telomeres and up-regulate *TERT* (telomerase reverse transcriptase), a gene located on chromosome 5 (5p15), conferring immortality to the mutated cells [18,24,41,42]. Deletion of *CDKN2A* (cyclin-dependent kinase inhibitor 2A), a tumor suppressor gene located on chromosome 9 (9p21), affects important pathways involved in cellular senescence [14,18,24,42]. Loss of the *PTEN* (phosphatase and tensin homolog) gene, located on chromosome 10 (10q23), up-regulates pro-survival pathways, such as PI3K/AKT1/mTOR [18,24,43]. Loss of *TP53* (tumor protein p53), a suppressor gene located on chromosome 17 (17p13), alters the mechanisms regulating cell and tissue homeostasis [18,24,44]. Other involved genes include *CDK4* (cyclin-dependent kinase 4), located on chromosome 12 (12q14), affecting pathways involved in cell cycle control [14,18,24,42], and chromatin-remodeling genes, such as *ARID1A* (AT-rich interactive domain-containing protein 1A) and *ARID1B* (AT-rich interactive domain-containing protein 1B), respectively, located on chromosomes 1 (1p36) and 6 (6q25) [18,24]. Further genomic alterations recently detected include a rare *RAF*-and phosphorylation-indipendent *MEK* (mitogen-activated protein kinase kinase 7—location 19p13.2) mutation [18,45] and amplification and overexpression of the gene *SEC62* (SEC62 homolog, preprotein translocation factor), located on chromosome 3q26 [18,46].

The majority of Spitz tumors, in association to the driver mutation/fusion, show no additional genomic abnormalities or only a few; a proportion of cases harbor a variable, relatively low, number of mutations; a fraction of them show a high number of genomic aberrations [13,38,40]. Therefore, considered as a whole, Spitz tumors seem to form a genomic spectrum, in which any given neoplasm shows a progressively increasing number of genetic alterations. In each lesion, the acquired tumor mutation burden (TMB) tends to generate a certain corresponding proportional clinical risk, that is, the probability of future possible adverse events (recurrences; local/regional/distant metastases; death) [40,47]. TMB is defined as the number of mutations seen in a section of DNA and reported as (mut/Mb), mutations per megabase (=1 million base) [47]. When TMB is low, constituted by the driver or a few more mutations, this risk is minimal, adverse events are rare or very rare, and there is a strong probability that, clinically and histologically, the tumor will be classified as “benign”. When TMB is high, the risk is high or very high, adverse events are relatively frequent, and clinically and histologically, the tumor will be probably classified as “malignant”. In cases with variable intermediate TMBs, the risk may range between a minimum and a theoretic maximum, and clinically and histologically, tumors will be probably classified as “intermediate”, “unpredictable” or “of uncertain malignant potential” [40].

## 4. Morphologic Aspects of Spitz Tumors

Morphologically, Spitz tumors are characterized by the presence of large spindle and/or epithelioid cells. However, they show some morphological differences, probably due to their different genetic drivers (Table 2).

*ALK*-fused Spitz tumors generally appear as amelanotic dome-shaped or polypoid lesions, located on the extremities of young individuals; histologically, they generally are prevalently dermal tumors, composed of plexiform fascicles of spindle cells, sometimes associated with epithelioid melanocytes showing prominent nucleoli [14,15,48,49]. *ROS1*-fused Spitz tumors are generally papular or dome-shaped lesions, most frequently on the lower extremities; microscopically, they generally appear as nodules, made up of large atypical epithelioid and spindle cells with vesicular nuclei and no specific cytological and histological features [14,15,50]. *NTRK*-fused Spitz tumors are generally symmetric exophitic/verrucuous compound or intradermal lesions; histologically, they show mildly or moderately pleomorphic spindle cells, with elongated rete ridges, lobulated dermal nests, rosette-like structures, exaggerated maturation and Kamino bodies [19,49,51]. *MET*-fused Spitz tumors do not seem to have characteristic histological features; they are generally symmetric, compound or intradermal dome-shaped lesions, showing epidermal hyperplasia and large nests of epithelioid or spindle cells [16]. *RET*-fused Spitz tumors also do not show typical morphologic characteristics, appearing as symmetric compound neoplasms with a plaque-like silhouette, showing large nests of epithelioid cells with mild-moderate nuclear atypia and cell dyscohesion [15,52]. *BRAF*-fused Spitz tumors are generally pink papular lesions, most frequently on the lower extremities of young patients; histologically, they mostly appear as intradermal nodular, plaque-like or wedge-shaped lesions, composed of moderately or severely atypical epithelioid and/or spindle melanocytes with vesicular nuclei and prominent nucleoli, arranged in dense cellular sheets and associated with dermal sclerosis. [15,49,53]. *MAP3K8*-fused Spitz tumors generally appear as pigmented asymmetric exophytic nodules, located on the lower extremities; microscopically, they generally are nodular, composed of prevalently severely atypical epithelioid cells, and sometimes multinucleated [31,54]. *HRAS*-mutated Spitz tumors generally appear as symmetrical nodular, plaque-like or wedge-shaped lesions, most frequently located on the head, neck or extremities; histologically, they are composed of epithelioid and/or spindle melanocytes with pleomorphic vesicular nuclei and ample cytoplasm, embedded in a desmoplastic stroma [27,55]. MAP2K1-mutated Spitz tumors generally are pigmented lesions of the lower extremities, frequently observed in young and female patients; microscopically, they are composed of large nests of epithelioid melanocytes with atypical vesicular nuclei, often in a plexiform arrangement, with convergence of nests around the adnexa and neurovascular bundles [29,56]. Spitz tumors occurring in acral sites appear as pigmented lesions, measuring 1–15 mm; histologically, they show large spindle and/or epithelioid cells, a lentiginous and nested intraepidermal component, extending beyond the dermal component, marked pagetoid spread, clefting and possible transepidermal elimination of junctional nests, Kamino bodies, melanin and melanophages in the papillary dermis, and sometimes a proliferation of dermal blood vessels [57,58,59].

In addition to these morphologic characteristics, Spitz tumors can show other histopathological features, reputed to be related with malignancy, such as marked asymmetry, poor circumscription, marked nuclear atypia, high number of mitotic figures, deep and/or marginal mitoses, extensive pagetoid spread, solid cellular sheets, epidermal ulceration, cellular necrosis and heavy inflammatory infiltrate [60]. All or many of these atypical features, generally associated with an aggressive biological behavior, can be found in a restricted fraction of Spitz tumors, while the majority of lesions exhibit none of these features, or only a limited number of them, and generally show a non-aggressive biological behavior. Therefore, considered as a whole, Spitz tumors have been viewed as forming a bio-morphological spectrum [13,32,38,61], caused by the progressive accumulation of chromosomal aberrations. In this bio-morphological spectrum, paralleling the genomic one, lesions show a progressive increase in level of clinical risk associated with a parallel progressive increase in cyto-architectural ‘deformation’ (“histological atypia”) [32,38,61]. Lesions progressively appear to show dimensional increase, due to the failure to stop the cellular growth; marked asymmetry, due to the development of cellular clones with a different replication speed; deep dermal or subcutaneous involvement, due to an enhancement of capability of growth; higher grade of cytological atypia, due to a progressive alteration of cell differentiation processes; cellular heterogeneity, due to the onset of tumor clones with new phenotypic characteristics; epidermal ulceration, due to an increase of destructive capacity; higher number of mitoses, due to the increase of proliferative activity; atypical mitoses, due to a progressive derangement of the cell replication process; possibly cells in lymphatic vessels, due to the acquisition of the capability of penetrating the vascular walls. This relationship, existing between biologic behavior and morphology, seems to be at the basis of the histological diagnosis, in which, analyzing the microscopic features of a given tumor, a clinical evaluation is possible [62].

Unfortunately, however, in Spitz tumors, the levels of clinical risk and histological atypia do not always seem to follow a parallel trend and this possible misalignment seems to contribute significantly to making histological diagnosis more difficult. Clinicopathologic correlations show that Spitz tumors tend to display a series of atypical histological features very early [4,61]. Lesions of recent onset may show features such as large epithelioid and/or spindle cells with enlarged and polymorphous nuclei, large multinucleated melanocytes, nuclear pseudo-inclusions, crowding of junctional nests and suprabasal cells [4,32,61,63]. However, these atypical features, entirely or in part, probable phenotypic effects produced by the initial driver fusion/mutation, appear associated with no significant clinical risk [4,32,61,63]. Moreover, clinicopathologic studies also show that the time may influence the clinical risk. In prepubertal patients and in adolescents, as well as in small lesions, a relevant histological atypia can be found, but clinical risk is low and the prognosis generally good [4,32,61], probably because the oncogenetic process has a high probability of being relatively recent and still in its early phases. On the other hand, in adult patients, as well as in large tumors, the risk level is higher and the prognosis less favorable [32,61,63,64], probably because the oncogenetic process is less recent and in more advanced phases, so that a higher number of mutations affecting the biological behavior have had more time to occur. Therefore, some Spitz tumors appearing as histologically markedly atypical may not be clinically aggressive, but susceptible to be overdiagnosed as Spitz melanomas. Conversely, some cases not necessarily displaying a particularly severe histological atypia may be clinically aggressive, but susceptible to be underdiagnosed as Spitz nevi [12,63,64]. In sum, histological misinterpretations, possible in both senses, do not seem necessarily due to an inappropriate use of the microscopic criteria or to a presumptive failure of the human brain [1], but may also depend on an actual lack of correspondence between morphologic features and clinical behavior [12].

## 5. A Retrospective Look at Ackerman’s Conundrum

The current landscape of melanocytic tumors appears far more complex than it did at the end of the last century. When Dr. Ackerman discussed the problem of differential diagnosis between Spitz nevus and melanoma, he considered Spitz nevus as a single distinct lesion, nothing more than one of the many morphologic variants of melanocytic nevus [65]. Similarly, accordingly to his “unifying concept of melanoma”, Dr. Ackerman viewed melanomas as a unique neoplasm, regarding its different histological appearances as pure morphological fluctuations, without real biological implications, due only to the specific characteristics of the anatomic sites in which it had arisen [66]. Certainly, there were no particular problems in the distinction of Spitz nevus from the most common histologic forms of melanomas, such as stereotypical superficial spreading melanomas, lentigo maligna melanomas or acral lentiginous melanomas. The real challenge was to distinguish Spitz nevus from epithelioid and spindle cell melanoma resembling Spitz nevus. Dr. Ackerman believed that these two lesions, having many histopathological features in common, were like two “identical twins”, but they could be distinguished by means of a list of histological criteria [1]. However, the problem was more complex, because Dr. Ackerman’s list was not the only proposed one, because several other diagnostic lists existed [64]. These lists contained a certain number of parameters to be assessed in any given lesion; the diagnosis had to emerge from a global evaluation of the listed features. However, unfortunately, the available diagnostic lists were different, did not contain the same number of parameters and did not always include the same criteria. It was also unclear if a given list was more effective than another. Moreover, no quantitative criteria to evaluate the given parameters had been defined; it had not been specified how many parameters were required for the diagnosis; if the presence of the majority of them was required or not; if all parameters had the same diagnostic weight or not; if there existed major and minor parameters and, in this case, how many major and how many minor parameters were necessary. Lastly, no indication for a qualitative evaluation of parameters was provided [67]. In this framework, the diagnosis could have been really problematic and the means proposed by Dr. Ackerman to solve his conundrum did not appear entirely conclusive.

The recent genomic acquisitions propose a different perspective. Spitz nevus is not a simple morphologic variant of melanocytic nevus and epithelioid and/or spindle cell melanoma (Spitz melanoma) is not a mere morphologic variant of melanoma. As previously hypothesized by evaluating cell morphology [68,69], the genomic studies have shown that Spitz tumors constitute an autonomous class of melanocytic lesions, because they harbor genetic drivers different from those harbored by other classes of melanocytic tumors [14,38]. Tumors belonging to this class do not really appear entirely homogeneous, because they show an assortment of different genetic drivers that, however, end up substantially activating the same signaling pathways [14,15]. Lesions diagnosed as Spitz nevi and lesions diagnosed as Spitz melanomas, both characterized by large spindle and/or epithelioid cells, are morphologically similar, but genomic studies suggest that it is not appropriate to regard their relationship as a matter of resemblance, imitation, or simulation [1,63,64]. Spitz nevus does not imitate or simulate Spitz melanoma; Spitz melanoma does not simulate or mimic Spitz nevus. They do not simulate each other and are not sly or deceiving tumors. There are no such things as lambs, sheep or wolves or other animals, often metaphorically invoked [70]. There is a class of large spindle and/or epithelioid cell tumors that show similar morphological and phenotypical characteristics, simply because they harbor the same genetic drivers, and show different biologic behaviors, simply because they possess different TMBs [14,38,40,70]. Actually, Spitz nevus and Spitz melanoma seem to be much more than identical twins. They can be regarded as lesions representing different evolutionary stages of the same oncogenetic pathway [38,40]. In this pathway, the accumulation of genomic aberrations is gradual and produces a bio-morphologic spectrum, comprising tumors that clinically and histologically show benign, intermediate and malignant characteristics [38,40]. Since this spectrum is continuous, a binary classification of lesions (“Spitz nevus—melanoma”) appears to be poorly adherent to the real neoplastic process, ignoring the many existing intermediate lesions. The need to include any single case either into the category “Spitz nevus” or into the opposite category “melanoma” or “Spitz melanoma” can induce a certain number of inappropriate diagnoses, especially when tumors show non-stereotypical characteristics, as not rarely occurs.

## 6. Risk-Associated Parameters in Spitz Tumors

The current histological WHO classification has incorporated several concepts derived by the genomic studies and subdivided Spitz neoplasms into “Spitz nevus”, “Spitz melanocytoma/atypical Spitz tumor” and “Spitz melanoma” [38]. This ternary classification is more adherent to the real neoplastic pathway and most tumors can be consistently diagnosed. However, lesions to be included into these three diagnostic categories, sharing a common driver mutational set and, consequently, the same pathogenesis, cannot be properly viewed as different tumors [40]. From a genomic point of view, the terms “Spitz nevus”, “Spitz melanocytoma/atypical Spitz tumor” and “Spitz melanoma” appear to be three diagnostic labels applied to three more or less defined collections of lesions, corresponding to three segments of the same oncogenetic process. Such segments are not well defined, because it is not easy to segment a continuous process into discrete entities. In fact, although progressively more severely atypical, the lesions to be classified under these three labels share many histological features and it may not be easy to establish the respective histological boundaries of the pre-set diagnostic categories in which they are to be included. Thus, for a number of cases, and especially for those with intermediate and potentially ambiguous features to be included in the category “Spitz melanocytoma/atypical Spitz tumor”, a precise diagnostic definition remains difficult [71]. This can produce a certain number of inappropriate diagnoses. Moreover, lesions potentially diagnosable as “Spitz melanocytomas/atypical Spitz tumors” could be more numerous than expected, because it seems relatively unlikely that mutations accumulate in large numbers rapidly. It seems more likely that a significant number of lesions may have an intermediate TMB, consequently appearing as potential candidates to be included in the intermediate diagnostic category. However, since Spitz melanocytomas/atypical Spitz tumors are currently reputed to represent only a small fraction [12,63], many tumors with intermediate features come to be forced into the “Spitz nevus” category or the opposite category of “Spitz melanoma”. This may contribute to produce a certain additional number of inappropriate diagnoses. Lastly, for the histopathological differential diagnosis of Spitz nevus, Spitz melanocytoma/atypical Spitz tumor and Spitz melanoma, a diagnostic list of histological features has been provided [72]. This list is accurate and detailed, but not devoid of problems; because parameters were evaluated comparatively, rather than in a binary way (present/absent), it was not specified how many parameters are required for diagnosis and whether all parameters had the same diagnostic weight or not. Therefore, although most cases can be diagnosed with relative confidence, for these and other possible unrecognized reasons, misinterpretations can occur. The common experience and the pertinent literature show that inappropriate diagnoses are possible, even when the proposed diagnostic criteria are appropriately applied. This is demonstrated by the unsatisfying level of interobserver agreement recorded in several old and recent studies, even among experts [12,60,73,74]. If diagnostic difficulties existed among experienced dermatologists, among less experienced pathologists, greater problems are reasonably to be expected.

Opportunely, problematic cases can be investigated with regard to the clinical risk they may confer on patients, by analyzing certain clinical, histopathological and genomic features (Table 3). Some clinical parameters, possibly related to the evolutionary stage of the oncogenetic process, such as advanced patient’s age and tumor size, seem to imply more clinical risk [60,75]. Moreover, some histological features, commonly used for the diagnosis, seem to be associated with an increased clinical risk and have been proposed as prognostic parameters [60,75,76]. A list of parameters to be used with a prognostic, rather than diagnostic, purpose may include features such as solid nodular growth, deep dermis and/or subcutaneous fat extension, marked nuclear pleomorphism/atypia, marked asymmetry, high number of suprabasal melanocytes, epidermal ulceration, dermal mitoses (≥2 per mm^2^), abundant melanin in deep cells, deep/marginal mitoses, cellular necrosis, cells in lymphatic vessels and heavy inflammatory infiltrate [60,75,76] (Figure 1 and Figure 2). In addition, other risk-associated parameters seem to come from genomic studies. The different genetic drivers do not seem to have a significant impact on the prognosis; however, Spitz tumors harboring *BRAF* and *MAP3K8* fusions, as well as *MAP3K8* truncations, seem to exhibit more aggressive clinical behavior [22,31,49,53,54]. Chromosomal abnormalities, such as a gain of 6p25 (*RREB1*, element 1 binding protein reactive to ras), a gain of 11q13 (*CCND1*, cyclin D1) and a deletion of 6q23 (*MYB*, v-myb myeloblastosis viral oncogene homologous) may indicate an increased clinical risk level [18,77,78]. The presence of a homozygous deletion of the *CDKN2A* gene (9p21) increases the risk of possible adverse events [77]. Cases showing a mutation of the *TERT* promoter show a less favorable prognosis [79]. A high TMB (>5 mut/Mb), measured by NGS, is associated with a less favorable biologic behavior [80]. In addition, tumors that exhibit histopathological features typical of Spitz tumors but harbor *BRAF*^V600E^, *NRAS* or *NF1* mutations often exhibit aggressive biological behavior and a less favorable outcome [24,34]. Cases with *BRAF* mutations (BAMS) [34], but probably also cases with *NRAS* and *NF1* mutations, have been regarded as possible subtypes of melanocytomas, that is, tumors with an intermediate malignant potential [81]. Moreover, re-proposing the simulation theory, it has been suggested that such neoplasms are not true Spitz tumors, but potentially aggressive melanocytic neoplasms that mimic them [24,37,81]. However, since the exact genetic structure of all Spitz tumors has not yet been exhaustively explored, and because these neoplasms appear to be morphologically indistinguishable from Spitz lesions, they could also be Spitz tumors harboring different and clinically more aggressive driver mutations. The integration of clinical and histopathological features with FISH analysis and genomic data by next-generation sequencing (NGS) analysis seems to improve the diagnosis and the interobserver agreement [82,83,84].

## 7. Conclusions and Future Directions

In conclusion, one of the reasons for Ackerman’s conundrum really seems to be related to the resemblance between Spitz nevus and melanoma [1]. However, this resemblance, rather than due to a deceitful reciprocal simulation, seems to be more simply due to the fact that both these tumors are part of the same neoplastic spectrum [70]. However, in the new scenario opened up by genomic studies, a deeper and more relevant reason for Ackerman’s conundrum seems to reside in the difficulty of diagnosing a spectrum of lesions using a binary classification (Spitz nevus-melanoma). A ternary classification (Spitz nevus-Spitz melanocytoma/atypical Spitz-Spitz melanoma) appears to be more adherent to the actual neoplastic pathway, because it provides for an intermediate category, which may correspond to the intermediate segment of the process. However, the “Spitz melanocytoma/atypical Spitz tumor” category does not appear to have well-defined histological borders and some cases remain difficult to diagnose [71]. Moreover, this terminology conveys diagnostic uncertainty, and predicting the clinical behavior of such ambiguous lesions can be challenging [14,71]. In general, neoplasms labeled as “Spitz melanocytomas/atypical Spitz tumors” harboring tyrosine kinase fusions rarely show adverse events such as distant metastasis and death [81]. Cases showing serine/threonine kinase fusions may be more aggressive, but the overwhelming majority of them seem to be indolent [81]. However, in a small fraction of these cases, although statistically unlikely, an adverse event is not impossible. The statistical incidence of these unfavorable events is low, but potentially harmful lesions should be identified. In major diagnostic centers, where a relatively large number of Spitz lesions are diagnosed each year, the incidence of such risky cases may not be negligible. In addition, the patient with a problematic lesion, associated with any adverse events, does not mind that the theoretical risk was statistically low. Therefore, all problematic cases should be studied as thoroughly as possible.

The possibility that cases can be further evaluated, by studying its clinicopathological and genomic characteristics, can also open a new perspective. Like problematic cases, all Spitz tumors could be prognostically assessed for their inherent clinical risk, that is, the probability of an adverse event occurring [40,75,76]. According to the TMB of each individual neoplasm, in fact, all Spitz tumors carry some level of risk, including lesions diagnosed as “Spitz nevi”, which also imply a risk, albeit very small [67,72,85]. The level of risk could be estimated as very low/minimal and clinically negligible, low, moderate, high or very high, and clinically relevant (Table 4).

Such prognostic stratification, complementary, if not alternative, to diagnosis and based on an integrated assessment of the morphological and genomic risk-associated features, could contribute to plan the optimal treatment of patients [75,76]. To this end, further advances to be achieved may include: (1) a comprehensive recognition of risk-significant chromosomal aberrations; (2) an estimation of the prognostic weight of each individual morphological and genomic parameter; (3) an evaluation of the possible unfavorable synergistic effects produced by selected collections of genomic abnormalities and of atypical morphological features, obtained by studying the follow-up data of the patients.

## Figures and Tables

**Figure 1 cancers-15-05834-f001:**
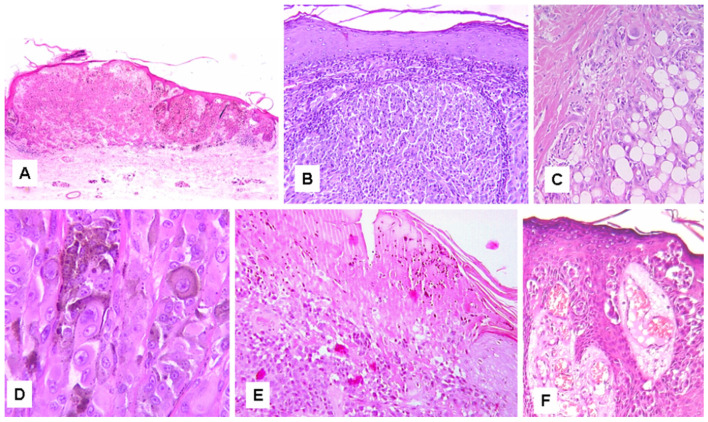
Risk-associated histopathological features in Spitz tumors (I). (**A**) Tumor asymmetry (hematoxylin and eosin stain, original magnification ×25). (**B**) Solid nodular growth (hematoxylin and eosin stain, original magnification ×100). (**C**) Deep dermis and subcutaneous fat extension (hematoxylin and eosin stain, original magnification ×200). (**D**) Marked nuclear pleomorphism/atypia (hematoxylin and eosin stain, original magnification ×400). (**E**) Epidermal ulceration (hematoxylin and eosin stain, original magnification ×125). (**F**) High number of suprabasal melanocytes (hematoxylin and eosin stain, original magnification ×100).

**Figure 2 cancers-15-05834-f002:**
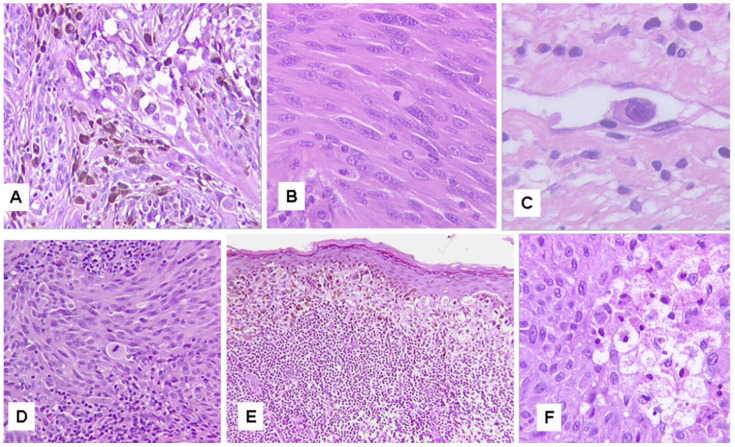
Risk-associated histopathological features in Spitz tumors (II). (**A**) Melanin in deep cells (hematoxylin and eosin stain, original magnification ×200). (**B**) Dermal mitotic figures (hematoxylin and eosin stain, original magnification ×400). (**C**) Cell in a lymphatic vessel (hematoxylin and eosin stain, original magnification ×400). (**D**) Mitosis in the deep portion of the tumor (hematoxylin and eosin stain, original magnification ×125). (**E**) Heavy inflammatory infiltrate (hematoxylin and eosin stain, original magnification ×100). (**F**) Focal cellular necrosis (hematoxylin and eosin stain, original magnification ×400).

**Table 1 cancers-15-05834-t001:** Genetic drivers in Spitz tumors.

Type of Alteration	Gene (HGNC Name)	Location	Partners Genes
Fusion	*ALK*	2p23	*TMP3*, *DCTN1*, *MLPH*
Fusion	*ROS1*	6q22	*PWWP2A*, *TMP3*
Fusion	*NTRK1*	1q23	*LMNA*, *TP53*, *TMP3*
Fusion	*NTRK2*	9q21	
Fusion	*NTRK3*	15q25	*ETV6*, *MYO5A*, *MYH9*
Fusion	*RET*	10q11	*GOLGA5*, *KIF5B*
Fusion	*MET*	7q31	
Fusion	*BRAF*	7q34	*CEP89*, *LSM14A*, *AKAP9*, *MAD1L1*
Fusion	*MAP3K8*	10p11	*SVIL*, *UBL3*, *SPECC1*, *STX7*, *GNG2*, *DIP2C*, *PRKACB*, *CUBN*
Fusion	*ARAF*	Xp11	
Fusion	*MITF*	3p14	
Fusion	*PRKCA*	17q24	
Fusion	*PRKCB*	16p12	
Fusion	*RASGRF1*	15q25	
Fusion	*PRKDC*	8q11	
Fusion	*FGFR1*	8p11	
Fusion	*ERBB4*	2q34	
Fusion	*RAF1*	3p25	
Fusion	*MAP3K3*	17q23	
Fusion	*MERTK*	2q13	
Fusion	*LCK*	1p35	*TRPM1*, *PUM1*
Mutation	*HRAS*	11p15	
Mutation	*MAP3K8*	10p11	
Mutation	*MAP2K1*	15q22	

HGNC = HUGO (Human genome organisation) Gene Nomenclature Committee; References [13,14,15,16,17,18,19,20,21,22,23,24,25,26,27,28,29,30,31,32]. For further details see text.

**Table 2 cancers-15-05834-t002:** Morphologic aspects of Spitz tumors.

Gene, Alteration	Clinical Features	Histopathological Features
*ALK*, fusion	Amelanotic dome-shaped or polypoid lesions, located on the extremities, young patients	Dermal tumors with plexiform fascicles of spindle cells, epithelioid melanocytes and prominent nucleoli
*ROS1*, fusion	Papular or dome-shaped lesions, frequently of the lower extremities	Nodules of large atypical epithelioid and spindle cells with vesicular nuclei; no specific histological features
*NTRK1-3*, fusion	Symmetric exophitic/verrucuous lesions	Compound or intradermal lesions with pleomorphic spindle cells, with elongated rete ridges, lobulated dermal nests, rosette-like structures, exaggerated maturation and Kamino bodies
*MET*, fusion	Symmetric, dome-shaped lesions	Compound or intradermal tumors with epidermal hyperplasia and large nests of epithelioid and/or spindle cells
*RET*, fusion	Symmetric plaque-like lesions	Symmetric compound lesions with large nests of epithelioid cells, mild-moderate nuclear atypia and cell dyscohesion
*BRAF*, fusion	Pink papular lesions on the lower extremities, young patients	Intradermal nodular, plaque-like or wedge-shaped lesions with cellular sheets of moderately or severely atypical epithelioid and/or spindle melanocytes, vesicular nuclei, prominent nucleoli and dermal sclerosis
*MAP3K8*, fusion or mutation	Pigmented asymmetric exophytic nodules on the lower extremities	Nodular lesions composed of severely atypical epithelioid cells, sometimes multinucleated
*HRAS*, mutation	Symmetrical nodular, plaque-like or wedge-shaped lesions, on head, neck or extremities	Nodular lesions composed of epithelioid and/or spindle melanocytes with pleomorphic vesicular nuclei, ample cytoplasm and desmoplastic stroma
*MAP2K1*, mutation	Pigmented lesions on the lower extremities, young and female patients	Tumors composed of large nests of epithelioid melanocytes with atypical vesicular nuclei, plexiform arrangement, convergence of nests around the adnexa and neurovascular bundles

References: [14,15,16,19,27,29,31,32,48,49,50,51,52,53,54,55,56].

**Table 3 cancers-15-05834-t003:** Clinical, histological and genomic risk-associated parameters in Spitz tumors.

Patient’s age > 40 year-old
Tumors size ≥ 10 mm
Solid nodular growth
deep dermis/subcutaneous fat extension
marked nuclear pleomorphism/atypia
marked asymmetry
high number of suprabasal melanocytes
epidermal ulceration
dermal mitoses ≥ 2 per mm^2^
deep/marginal mitoses
abundant melanin in deep cells
cellular necrosis
cells in lymphatic vessels
heavy inflammatory infiltrate
*BRAF* fusion
*MAP3K8* fusion/truncation
6p25 (*RREB1*) gain
11q13 (*CCND1*) gain
6q23 (*MYB*) loss
9p21 (*CDKN2A*) biallelic loss
TERT promoter mutation
High TMB > 5 mut/Mb
*BRAF* mutation ^°
*NRAS* mutation ^
*NF1* mutation ^

References [22,24,31,34,49,53,54,60,75,76,77,78,79,80]. ^ not considered true Spitz tumors (References [34,37]). ° provisionally included in BAMS category (Reference [34]). For further details see text.

**Table 4 cancers-15-05834-t004:** Possible prognostic stratification of in Spitz tumors.

ST1—Spitz tumor with minimal/very low risk	No risk-associated parameters
ST2—Spitz tumor with low/moderate risk	One or more risk-associated parameters
ST3—Spitz tumor with high/very high risk	One or more risk-associated parameters, including one or more of the following: mitoses ≥ 4/mm^2^, deep/marginal mitoses, heavy inflammatory infiltrate, biallelic deletion of 9p21 (*CDKN2A*), *MAP3K8* fusion/truncation, *TERT*-p mutation, high TMB, *BRAF/NRAS/NF1* mutations

References [22,24,31,34,49,53,54,60,75,76,77,78,79,80].

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
