# Peer review of "Spitz Tumors and Melanoma in the Genomic Age: A Retrospective Look at Ackerman’s Conundrum"

_cancers, 2023, doi:10.3390/cancers15245834_

Round 1

Reviewer 1 Report

Comments and Suggestions for Authors

The manuscript is well drawn and optimally documented. I suggest acceptance  after the following minor revisions:

-2 Genetic drivers in Spitz tumors: line 101. I suggest to modify the sentence “including LMNA...and TPM3” with “including LMNA..., TMP3, ETV6, MYO5A and MYH9” and cite the references doi:10.1038/s41379-020-00678-6, doi:10.3390/ijms222212332.

-4 Morphological aspects of Spitz tumors: line 235. Please cite the reference doi:10.1038/s41379-020-00678-6.

Author Response

General Comment: The manuscript is well drawn and optimally documented.

Response: I thank you for the encouraging comment and evaluation.

Comments 1: -2 Genetic drivers in Spitz tumors: line 101. I suggest to modify the sentence “including LMNA...and TPM3” with “including LMNA..., TMP3, ETV6, MYO5A and MYH9” and cite the references doi:10.1038/s41379-020-00678-6, doi:10.3390/ijms222212332.

Response 1: Thank you for pointing this out. I agree with this comment. I have modified the sentence, adding the suggested partner genes (2 - Genetic drivers, page 3). The relative references have been added.

Comments 2: -4 Morphological aspects of Spitz tumors: line 235. Please cite the reference doi:10.1038/s41379-020-00678-6.

Response 2: Agree. The suggested reference has been added.

Reviewer 2 Report

Comments and Suggestions for Authors

The paper presents a remarkable retrospective perspective, based on literature reports, concerning the morphological spectrum of Spitz neoplasms-SNs (tumors and melanomas) paralleling the (tendency of) genomic oncogenetic pathway spectrum, which (putatively) indicates a spectrum of clinical risk (aggressiveness behavior).

As the author claims, this updated reconsideration perspective may (prospectively) impact the optimal treatment of patients.

The time of SN evolution in each patient also impacts the clinical risk and, consequently, the management decision. The author might clarify this issue.

Also, he might briefly address (in a new Table, with minimal criteria/level of clinical risk) the practical implications of the proposed (probabilistic) prognostic (risk-associated) stratification in the management of patients with SNs.

The paper may turn out interesting to readers.

Author Response

General comment: The paper presents a remarkable retrospective perspective, based on literature reports, concerning the morphological spectrum of Spitz neoplasms-SNs (tumors and melanomas) paralleling the (tendency of) genomic oncogenetic pathway spectrum, which (putatively) indicates a spectrum of clinical risk (aggressiveness behavior). As the author claims, this updated reconsideration perspective may (prospectively) impact the optimal treatment of patients.

Response: I thank you for this clarifying comment.

Comments 1: The time of SN evolution in each patient also impacts the clinical risk and, consequently, the management decision. The author might clarify this issue.

Response 1: I agree. The issue is important, but still insufficiently studied. However, in the text, it was briefly discussed in 4 - Morphologic aspects, in which it was underlined that recent tumors (pediatric/young patient - small lesions) tend to have a less aggressive behavior than older ones (old patients - large lesions); this probably because mutations affecting the biological behavior could have had more time to occur (page 8). Following your suggestion, in order to better elucidate this point, the sentences regarding the possible relationship between time and clinical risk have been modified (page 8). Moreover, in "6-Risk-associated parameters", the sentence, regarding the possible relationship time-evolutionary stage of the neoplastic process-clinical risk, was modified (page 10).

Comments 2: Also, he might briefly address (in a new Table, with minimal criteria/level of clinical risk) the practical implications of the proposed (probabilistic) prognostic (risk-associated) stratification in the management of patients with SNs. The paper may turn out interesting to readers.

Response 2: The suggestion is really interesting. Currently, a definitive stratification of cases appears to be difficult, because the "exact" prognostic weight of the risk-associated parameters is not entirely clarified. Moreover, numerous studied daily emerge with new findings, modifying the general landscape. However, following your suggestion, a new Table (Table 4) was added, taking into account of some previous attempts of stratifications (refs. #75-76). I thank you for suggestion.

Reviewer 3 Report

Comments and Suggestions for Authors

Carmelo Urso described a review about genetic features of Spitz tumors. The content is interesting and important to understand development of Spitz nevus and Spitz melanoma.

My comments are as follows.

Please clearly describe the threshold of low TMB and high TMB.

It is better to make a Table summarizing morphologic aspects of Spitz tumors (4).

What is the frequent mutations in pediatric lesions?

Does the frequency of the mutation in Spitz tumor vary depending on body sites?

Spitz nevus rarely develops in acral lesions as these publication (PMID: 29537991, PMID: 21747632, PMID: 32096247). Author should add descriptions about Spitz nevus in acral lesion with adding some articles including PMID: 29537991, PMID: 21747632, PMID: 32096247.

There are other interesting articles such as PMID: 33669371, PMID: 33915997 and PMID: 35653096 related with this review. Author should add these articles to the reference with descriptions in main text for making more comprehensive review.

Author Response

General Comment: Carmelo Urso described a review about genetic features of Spitz tumors. The content is interesting and important to understand development of Spitz nevus and Spitz melanoma.

Response: I thank you for this nice comment.

Comment 1: Please clearly describe the threshold of low TMB and high TMB.

Response 1: TMB is reported as (mut/Mb), mutations per megabase (=1 million base) (JAMA Oncol. 2021; 7:316). It is not exhaustively studied in Spitz tumors. An approximate threshold for high TMB, as a possible risk-associated parameter, was indicated as 5 mut/Mb (6 - Risk-associated parmaters, page 10), because, in a series of Spitz tumors, cases diagnosed as melanomas have showed an average of 5.7 mutations/Mb (Pediatr Dermatol 2022; 39: 409-419).

Comment 2: It is better to make a Table summarizing morphologic aspects of Spitz tumors (4).

Response 2:  A Table summarizing morphologic features of Spitz neoplasms was added (Table 2).

Comment 3: What is the frequent mutations in pediatric lesions?

Response 3: In Spitz tumors (AST/SM) occurring in pediatric patients (<18yo), ALK and MAP3K8 fusions seem to be the most common genetic drivers (Cancer 2021;127:3825-3831). An apposite sentence has been added in the text (2 - Genetic drivers, page 4). I thank you for your suggestion.

Comment 4: Does the frequency of the mutation in Spitz tumor vary depending on body sites?

Response 4: I agree.  ALK and ROS1 fusions seem to be very  frequently found in tumors of the lower extremities. NTRK and BRAF fusions seem to prefer lesions of lower and upper extremities. HRAS mutation is more frequently found in tumors of the head-neck region, with the most frequent secondary location on the extemities. MAP3K8 fusion/truncation has been found in lesions of the extremities, trunk and head-neck region. MAP2K1 mutation has been observed in tumors located on the extremities (Virchows Archiv 2012; 479:195-202; AJSP 2019; 43:1631-1637; Front Oncol 2022; 12: 889223). Thank you for suggestion. An apposite sentence has been added in the text (2 - Genetic drivers, page 4).

Comment 5: Spitz nevus rarely develops in acral lesions as these publication (PMID: 29537991, PMID: 21747632, PMID: 32096247). Author should add descriptions about Spitz nevus in acral lesion with adding some articles including PMID: 29537991, PMID: 21747632, PMID: 32096247.

Response 5: I thank you for your suggestion. A description of acral Spitz nevus/tumors and the suggested references have been added (4 - Morphologic aspects, page 7).

Comment 6: There are other interesting articles such as PMID: 33669371, PMID: 33915997 and PMID: 35653096 related with this review. Author should add these articles to the reference with descriptions in main text for making more comprehensive review.

Response 6: I thank you for your suggestion. The suggested papers have been cited in the text  (3 - Other genomic aberrations, page 5; and in  6- Risk-associated, page 10). The relative references were added.

Reviewer 4 Report

Comments and Suggestions for Authors

A manuscript by Urso is an interesting review on Spitz tumors. The Author discusses current knowledge on this type of cancers with regard to up-to-date point of view, including e.g., genetic background. The manuscript is well and clearly written.

Specific comments:

1. Gene names should be written in italics.

2. Table 2 should be changed to e.g., a figure. It does not add any substantial value as a table. As the manuscript lacks any graphical elements, the Author should consider providing any figure to improve the quality of the review.

Author Response

General comment: A manuscript by Urso is an interesting review on Spitz tumors. The Author discusses current knowledge on this type of cancers with regard to up-to-date point of view, including e.g., genetic background. The manuscript is well and clearly written.

Response: I thank you for this nice encouraging comment.

Comments 1: Gene names should be written in italics.

Response 1: I agree. It is correct and I apologize. The text has been reviewed and gene names written in italics.

Comments 2: Table 2 should be changed to e.g., a figure. It does not add any substantial value as a table. As the manuscript lacks any graphical elements, the Author should consider providing any figure to improve the quality of the review.

Response 2: I have provided Figures 1 and 2, illustrating the risk-associated histological parameters. However, I would propose to keep Table 2 (now renumbered as Table 3), because it can be useful for the subsequently added Table (Table 4). I thank you for the suggestion.

Round 2

Reviewer 3 Report

Comments and Suggestions for Authors

The manuscript was significantly improved, and I do not have any more comments.